# Exosomes from Adipose-Tissue-Derived Stem Cells Induce Proapoptotic Gene Expression in Breast Tumor Cell Line

**DOI:** 10.3390/ijms25042190

**Published:** 2024-02-12

**Authors:** Oliver Felthaus, Simon Vedlin, Andreas Eigenberger, Silvan M. Klein, Lukas Prantl

**Affiliations:** 1Department for Plastic, Hand & Reconstructive Surgery, University Hospital Regensburg, 93053 Regensburg, Germanysilvan.klein@ukr.de (S.M.K.); lukas.prantl@ukr.de (L.P.); 2Medical Device Lab, Faculty of Mechanical Engineering, Ostbayerische Technische Hochschule Regensburg, 93053 Regensburg, Germany

**Keywords:** breast cancer, mammary carcinoma, lipofilling, autologous fat grafting, adipose-tissue-derived stem cells, exosomes, extracellular vesicles

## Abstract

Lipofilling is an option for breast reconstruction after tumor resection to avoid the complications of an implant-based reconstruction. Although some concerns exist regarding the oncological safety of tissue rich in mesenchymal stem cells with their proangiogenic and proliferation-supportive properties, there are also reports that adipose-tissue-derived stem cells can exhibit antitumoral properties. We isolated primary adipose-tissue-derived stem cells. Both conditioned medium and exosomes were harvested from the cell culture and used to treat the breast cancer cell line MCF-7. Cell viability, cytotoxicity, and gene expression of MCF-7 cells in response to the indirect co-culture were evaluated. MCF-7 cells incubated with exosomes from adipose-tissue-derived stem cells show reduced cell viability in comparison to MCF-7 cells incubated with adipose-tissue-derived stem-cell-conditioned medium. Expression of proapoptotic genes was upregulated, and expression of antiapoptotic genes was downregulated. The debate about the oncological safety of autologous fat grafting after tumor resection continues. Here, we show that exosomes from adipose-tissue-derived stem cells exhibit some antitumoral properties on breast cancer cell line MCF-7.

## 1. Introduction

Breast cancer is the most common malignant tumor in women with an incidence of around 25% and is responsible for 15% of all cancer-related deaths in women [1]. Whereas the mortality rates of breast cancer are declining due to improved therapeutic options and early detection screening, the incidence is continuously increasing [2,3]. Although nonsurgical therapy options including radiotherapy and chemotherapy are available, tumor resection remains the crucial mainstay [4,5], and for later stages of breast cancer, mastectomy is often inevitable [6]. This procedure is accompanied by a high degree of psychological strain for the patients, leading to a strong desire for breast reconstruction [7]. Until a few decades ago, implant-based reconstruction was the gold standard for breast reconstruction [8,9,10,11]. However, using the patients’ own tissue in a more regenerative approach has increasingly gained interest in the last years.

Lipofilling or autologous fat grafting (AFT) is one of the most common procedures in plastic surgery. Adipose tissue can be harvested with minimally invasive methods and is rich in adipose-tissue-derived stem cells (ADSCs). The immunomodulatory, angiogenic, and proliferation-supporting properties of ADSCs makes adipose tissue an ideal material for regenerative medicine [12,13,14]. Due to complications with silicone implants, AFT has also gained increasing interest as therapy after breast tumor resection and mastectomies in the last years [8]. In contrast to the implant-based reconstruction, the autologous reconstruction prevents the risk of implant failure or foreign body reactions. Additionally, autologous reconstruction leads to a higher patient satisfaction and postoperative quality of life [7]. When reconstruction with autologous tissue alone is not possible, for example in very lean patients, the additional usage of adipose tissue in the implant-based reconstruction has been shown to improve the aesthetic outcome [15,16]. However, there are some concerns regarding the oncological safety of autologous fat grafting. ADSCs may be capable of supporting neoangiogenesis and cell proliferation through paracrine signaling, and therefore, ADSCs may contribute to tumor relapse [17]. On the other hand, antitumoral properties of both ADSCs and extracellular vesicles from ADSCs have been reported [18,19].

The influence of cells on each other both between the same cells and different cell types often relies on paracrine signaling by secreted particles and molecules [20]. Extracellular vesicles are defined as naturally secreted particles with a lipid bilayer that are not self-replicatory [21]. Several subtypes can be identified that differ in terms of particle size as well as the composition of lipids, proteins, and nucleic acids [22]. The most important subtypes are microvesicles and exosomes. Microvesicles are between 100 nm and 1000 nm in diameter and, as ectosomes, emerge from the outward budding and fission of the plasma membrane. Exosomes (between 30 nm and 100 nm in diameter) are initially formed by endocytosis as endosomes. After combining with a multivesicular body, the endosomes are released as exosomes by exocytosis. Exosomes carry the tetraspanin cluster of differentiation 63 (CD63) on the surface. Exosomes have been shown to influence cellular behavior and might mediate the adaptive immune responses to tumors [23].

Apoptosis, the programmed cell death, is often inhibited in malignant cells. Apoptosis can be initiated by an intrinsic pathway after detection of cellular stress or by an extrinsic pathway activated by paracrine signaling [24]. However, the intrinsic pathway can also be activated by extracellular signaling [25]. In both cases, the activation of proteases from the caspase family of proteins results in cell death. Key players for the intrinsic pathway are proteins of the B-cell lymphoma 2 (Bcl-2) family. A translocation of Bcl-2-Associated X Protein (Bax) or Bcl-2 Homologous Antagonist/Killer 1 (Bak1) into the mitochondrion results in the destabilization of the mitochondrial membrane and the activation of caspases [24]. Bcl-2, on the other hand, stabilizes the mitochondrial membrane and inhibits apoptosis but is itself negatively regulated by Bcl-2-associated agonist of cell death (Bad) [26]. Apoptosis is often related to cell cycle arrest and the activation of p53 or RB1 and their proapoptotic roles. Activation of caspase 3 (Casp3) is one of the last steps during the apoptotic cascade in both the intrinsic and extrinsic pathways.

The proangiogenic and proliferation-supporting properties of the growth factors and cytokines secreted by the cells from the adipose tissue raise concerns about the oncological safety of lipofilling after tumor resection. On the other hand, exosomes have been shown to induce apoptosis [27]. Therefore, a negative impact of autologous fat grafts on residual tumor cells is possible, too. Here, we evaluated the effect of exosomes isolated from ADSCs on the viability and gene expression of breast tumor cell line MCF-7 in comparison to the conditioned medium (CM).

## 2. Results

### 2.1. Exosome Detection

Isolated exosomes were evaluated using Western blot. After electrophoretic separation and blotting, a signal from the mouse anti-human CD63 antibody can be detected between 30 kDa and 50 kDa. No signal is seen for the CM (Figure 1).

### 2.2. Cell Viability and Cytotoxicity

Cell viability was tested using a resazurin assay for MCF-7 cells treated with conditioned medium and medium supplemented with exosomes (ES) in the concentrations 10 µL/cm^2^, 20 µL/cm^2^, and 30 µL/cm^2^. Normal growth medium served as control. In comparison to the conditioned medium, the ES (exosome-supplemented) medium shows a concentration-dependent negative effect on MCF-7 cell viability. However, in the tested concentrations, no significant negative effect in comparison to the control medium can be seen (Figure 2). ADSC-conditioned medium has a positive effect on MCF-7 cell viability. In comparison to the conditioned medium, the ES medium in all tested concentrations and the control medium show a significantly reduced cell viability on day 2 and day 3 (*p*-value < 0.05).

Additionally, cytotoxicity was tested using a lactate dehydrogenase (LDH) assay for MCF-7 cells treated with conditioned medium and medium supplemented with exosomes in the concentrations 10 µL/cm^2^, 20 µL/cm^2^, 30 µL/cm^2^, and 35 µL/cm^2^. Normal growth medium served as control. The MCF-7 cells treated with ES medium show an increased LDH release after 24 h, both in comparison with the conditioned medium and the control medium. LDH release in the conditioned medium was lower than in the control medium, whereas LDH release was higher in all exosome concentrations than in the control medium (Figure 3). However, only the 35 µL/cm^2^ exosome concentration sample was significantly higher than the control medium.

### 2.3. Gene Expression

Real-time RT-PCR was performed to evaluate gene regulation after exosome treatment. Expression of pro-apoptotic genes TP53, Bax, Bad, Casp3, Bak1, and RB1 was generally higher in cells treated with the exosome medium compared to both the cells treated with the conditioned medium and the control cells. Only pro-apoptotic gene Bak1 was downregulated in exosome-treated cells compared to the control. For anti-apoptotic gene Bcl-2, angiogenesis-promoting genes vascular endothelial growth factor A (VEGF-A), platelet-derived growth factor A (PDGF-A), and platelet-derived growth factor B (PDGF-B), and the proliferation marker Ki67, expression was higher in cells treated with the conditioned medium compared to the exosome medium. However, except for PDGF-B, expression of these genes was not downregulated in exosome-treated cells in comparison to the control cells (Figure 4).

## 3. Discussion

Lipofilling is commonly used in plastic surgery for the treatment of soft tissue deficits [28,29,30]. Adipose tissue is rich in mesenchymal stem cells which possess a multipotent differentiation potential and have immunomodulatory, angiogenetic, and proliferation-supporting properties, making it an ideal biomaterial for regenerative medicine [12,13,31,32,33]. However, these properties might also exert a prooncogenic effect on residual cells after tumor resection. Therefore, AFT is viewed critically for breast reconstruction after mastectomy [34,35]. On the other hand, antitumor properties of mesenchymal stem cells have been reported. Especially, exosomes as part of the adaptive immune response to tumors could exert antitumoral properties [19,23].

The gold standard for exosome isolation is ultracentrifugation [36]. Lacking the necessary equipment, we have utilized a method based on magnetic beads. Exosomes were successfully isolated from ADSC-conditioned cell culture medium, as confirmed with immunodetection (Figure 1). CD63 is a marker for exosomes, although it has also been detected on the surface of other vesicles [37,38]. However, when separated electrophoretically, exosomes show a size range between 30 kDa and 60 kDa [39,40], matching the size of the particles detected with our Western blot.

Conditioned medium is rich in cytokines and growth factors. It has been shown that ADSC-conditioned medium can increase proliferation in a mammary carcinoma cell line [41]. This is in accordance with our findings. Compared to the control medium, ADCS-conditioned medium supported MCF-7 cell proliferation significantly (Figure 2). Additionally, ADCS-conditioned medium causes a reduced LDH release in MCF-7 cells, which suggests protective properties of ADSC from cell death (Figure 3). For the cytotoxicity assay, ES medium had the adverse effect. LDH release was higher for all tested exosome concentrations than in the control medium. However, only for the highest concentration these differences were statistically significant. In the viability assay, the effects of the ES medium were unambiguous. For all concentrations, the cell viability in ES-treated cells was lower than in the cells treated with CM, but only for the highest concentration cell viability was lower compared to the control medium, and this difference was statistically not significant. However, the shown viability decrease is concentration-dependent, and it is likely that with higher exosome concentrations, a more distinct impact on MCF-7 cell viability would have been observed. This is in accordance with the observation that the only significant LDH release increase was seen with the 35 µL/cm^2^ concentration, which was not tested in the viability assay. Preparation of exosome isolate is material- and time-consuming. The LDH assay showed significant results after 24 h. Therefore, fewer samples were needed, which allowed for the testing of higher exosome concentrations. For the viability assay, new ES medium was needed after every measurement, restricting the examinable concentrations. However, for following studies, a higher concentration should be used for the viability assays.

There are several properties a cell needs to acquire to become a tumor cell. Among these hallmarks of cancer are the ability to evade apoptosis as well as sustained angiogenesis [42,43]. Therefore, it is important whether ADSCs support angiogenesis and apoptosis resistance in residual tumor cells at the recipient site after AFT. ADSC CM enhances the gene expression of genes for PDGF and VEGF, which are important for neovascularization, of the proliferation marker Ki67, and the antiapoptotic gene Bcl-2 both in comparison to the control medium and the ES medium. Antiapoptotic genes on the other hand are downregulated in comparison to the ES medium. In contrast, MCF-7 cells treated with ES medium showed increased gene expression for pro-apoptotic genes compared to both cells treated with the control medium and cells treated with ADSC CM. This is in accordance with other studies where a pro-apoptotic effect of exosomes isolated from mesenchymal stem cells was shown [44]. In our study, antiapoptotic genes, proliferation marker, and angiogenesis-related genes are downregulated in comparison to cells treated with ADSC CM. However, compared with the control medium, these markers were upregulated, also for the tested concentration. The strong increase in LDH release observed for the highest exosome concentration supports the hypothesis that a higher exosome concentration could cause a further downregulation of these genes, resulting in increased apoptosis and decreased proliferation and angiogenesis. However, it is also possible that a more distinct gene regulation is not necessary to explain the pronounced effect observed for the LDH assay. An important factor for the intrinsic pathway of apoptosis initiation is the release of cytochrome c after translocation of pro-apoptotic regulators Bad or Bax across the mitochondrial membrane [26], which leads to the activation of caspases and eventually to cell death [45]. Bcl-2, on the other hand, stabilizes the barrier function of the mitochondrial membrane and inhibits the translocation across the membrane and therefore the release of cytochrome c [46]. Therefore, for the initiation of apoptosis, the ratio between Bcl-2 and Bad is important. If Bcl-2 is predominant, apoptosis is inhibited; if Bad is predominant, apoptosis is induced [47]. This mechanism might explain why small changes in the relative gene expression of these genes after CM or ES medium incubation can cause a profound difference in apoptosis induction observed with the LDH assay. Although antiapoptotic Bcl-2 is upregulated in both CM and ES medium, the cells in the ES medium might be shifted towards apoptosis because Bad and Bax are upregulated as well. The Bcl-2 upregulation, which is less profound than in the CM, is more than compensated by the Bad- and Bax-upregulation, shifting the ratio to an apoptosis-inducing level. Interestingly, Bad is also connected to the inhibition of metalloproteases and was shown to interfere with the epithelial–mesenchymal transition, which is needed for tumor metastasis [48,49].

Besides apoptosis evasion, the bypassing of tumor suppressor mechanisms including cell cycle arrest is an important hallmark of cancer [42]. The tumor suppressor genes TP53 and RB1 are among the ones most often found mutated in tumors [50,51]. However, the MCF-7 cell line is reported to be both TP53 and RB1 wild type [50,52]. Therefore, the upregulation seen in the PCRs can be expected to be of biological relevance. This is in accordance with the observed cell vitality in the corresponding 30 µL/cm^2^ concentration, where cell vitality is not increasing for the evaluated period of time, suggesting a possible cell cycle arrest. In the conditioned medium, however, the very small and not significant upregulation of TP53 and RB1 might not be sufficient to have observable effects on cell viability.

The growth factors and cytokines found in ADSC CM support cell proliferation, cell survival, and angiogenesis. Therefore, concerns about the oncological safety of AFT after tumor resection are justified. However, in our own studies, we did not find any evidence that autologous fat grafting increases the risk for cancer recurrence [53]. Here, we confirm these properties for the ADSC secretome. However, in most cases of lipofilling, these are desired properties to improve the graft take rate. Actually, efforts are being made to improve transplant survival through enhanced angiogenesis and cell survival. In order to reach this goal, lipoaspirate is for example enriched with stem cells or supplemented with platelet-rich plasma [54,55]. Nonetheless, the risk of tumor recurrence from the ADSC secretome cannot be dismissed. Exosomes isolated from ADSCs, however, seem to have a concentration-dependent opposite effect in regard to apoptosis, proliferation, and angiogenesis. Perhaps the intentional enrichment of lipoaspirate with exosomes prior to injection for tissue augmentation should be considered specifically for post-oncologic patients, just like adipose tissue lipoaspirate is currently augmented with growth factors or growth-factor-producing stem cells for non-oncologic indications.

One limitation of our study is the exosome isolation method without ultracentrifugation. In future studies, the exosome concentration that caused a significant increase in LDH increase or even higher concentrations should be tested for their effect on cell viability and gene expression. Additionally, the characterization of exosomes with CD63 only should be extended to more markers in future studies.

## 4. Materials and Methods

### 4.1. Cell Culture

ADSCs were isolated as previously described [56]. Briefly, lipoaspirate from patients who underwent liposuction was enzymatically digested with a 0.05% collagenase solution (Collagenase from *C. histolyticum*, Sigma-Aldrich, St. Louis, MO, USA) at 37 °C for 60 min, filtered through a 100 µm filter, and centrifuged at 500 rcf for 5 min. The supernatant was discarded, and the cell pellet was washed with PBS (Pan Biotech, Aidenbach, Germany) and seeded into cell culture flasks with αMEM (Pan Biotech) supplemented with 10% fetal calf serum and 1% of a penicillin/streptomycin solution (growth medium). When reaching subconfluency, cells were detached using a trypsin/EDTA solution and seeded in a density of 5000 cells per cm^2^. Cells are routinely evaluated for surface marker expression (CD44^+^, CD73^+^, CD90^+^, CD105^+^, CD45^+^) and differentiation potential. ADSCs in passage 3 were used for all experiments. MCF-7 cell line was purchased from CLS Cell Lines Service (Eppelheim, Germany) and cultured in the same growth medium. Seeding density was 30,000 cells per cm^2^.

### 4.2. Medium Harvesting, Exosome Isolation, and Exosome Detection

ADSCs were seeded into cell culture flasks. Upon reaching subconfluency, the medium was changed to a serum-free medium. The medium was harvested 24 h later. For exosome isolation, the harvested medium underwent serial centrifugation (300 rcf for 10 min, 2000 rcf for 20 min, and 4500 rcf for 45 min) to remove cell debris. The supernatant was transferred to a new centrifugation tube after every centrifugation step and filtered through a 0.2 µm filter after the last centrifugation step. Subsequently, the exosomes were preconcentrated using Centricon Plus-70 Centrifugal Filter Units (Merck Millipore, Burlington, MA, USA) with a Molecular Weight Cut-Off of 100 kDa. In this way, a quantity of 75 mL of medium collected from 5 T175 cell culture flasks with a combined growth area of 875 cm^2^ was concentrated to a volume of ca. 350 µL. For the final exosome isolation from the medium concentrate, the Pan Human Exosome Isolation Kit (Miltenyi Biotec, Bergisch Gladbach, Germany) based on magnetic-beads-coupled antibodies was used according to the manufacturers’ instructions. The final isolation volume was 100 µL. For the experiments, normal growth medium was supplemented with different concentrations of exosome isolate. For standardization, the volume of exosome isolate per treated cell growth area in µL/cm^2^ was used as unit.

ADSC-conditioned medium was also harvested after 24 h incubation in serum-free medium and was supplemented with 10% fetal calf serum prior to the experiments.

Isolated exosomes were detected using Western blot. For SDS-PAGE, the Mini-PROTEAN^®^ Tetra Cell electrophoresis chamber (BioRad, Hercules, CA, USA) and the PageRuler Prestained NIR Protein Ladder (Thermo Fisher Scientific, Waltham, MA, USA) were used. After electrophoresis, separated vesicles were blotted onto a nitrocellulose membrane (Amersham Protran 0.2 NC, Sigma-Aldrich) using the Mini Trans-Blot^®^ Cell wet-blotting system (BioRad). Membranes were incubated with primary antibody (mouse anti-human CD63 antibody (Thermo Fisher Scientific)) overnight at 4 °C and the next day with infrared-labeled secondary antibody IRDye 680CW goat anti-mouse (Li-Cor Biosciences, Lincoln, NE, USA) for one hour at room temperature. Visualization was performed using the Odyssey^®^ Infrared Imaging System.

### 4.3. Viability and Cytotoxicity Assays

MCF-7 cells were seeded into 96-well plates (4800 cells/cm^2^) and incubated with different concentrations of ES medium, with CM and normal growth medium as control. Viability was evaluated using a resazurin assay on 4 consecutive days, where day 0 refers to the measurement prior to the start of treatment. Cells were incubated with normal growth medium supplemented with 0.07 µM of resazurin (Sigma-Aldrich) for two hours. Metabolic conversion of resazurin into the fluorescent resorufin was detected using a multiwell plate reader (excitation: 530 nm, emission: 590 nm, Varioskan, Thermo Fisher Scientific). Cytotoxicity was evaluated using the LDH Assay (abcam, Cambridge, UK) according to the manufacturers’ instructions.

### 4.4. RNA Isolation and RT-PCR

MCF-7 cells were seeded into 24-well plates (50,000 cells per well) in triplicates and allowed to adhere. Afterwards, the medium was changed to control medium (normal growth medium), CM, or ES medium (30 µL/cm^2^). RNA was harvested after 24 h using the RNeasy Mini Kit (Qiagen, Hilden, Germany) according to the manufacturer’s instructions. Isolated RNA was reversely transcribed using the QuantiTect Reverse Transcription Kit (Qiagen). For real-time RT-PCR, the DyNAmo HS SYBR Green qPCR Kit (Life Technologies, Carlsbad, CA, USA) with the Eco™ Real-Time PCR System (Illumina, San Diego, CA, USA) was used. Primers were designed using Primer3 and were purchased from Eurofins MWG Operon (Ebersberg, Germany). Primer sequences are listed in Table 1. The relative gene expressions were calculated using the ΔΔC_t_ method and normalized to the control using glycerinaldehyd-3-phosphat-dehydrogenase (GAPDH) as the housekeeping gene. The gene expression was evaluated for genes related to angiogenesis and apoptosis.

### 4.5. Statistical Analysis

For statistical analysis of viability and cytotoxicity assays, the Student’s *t*-test was used. For PCR analysis, ANOVA was utilized. Normal distribution was assessed using the Kolmogorov–Smirnov test. Variance homogeneity was evaluated using Levene’s test. If homogeneity of variance is given, Bonferroni was used for post hoc test, and Dunnett T3 was used for post hoc test when homogeneity of variance was not given. *p*-values below 0.05 were considered statistically significant.

## Figures and Tables

**Figure 1 ijms-25-02190-f001:**
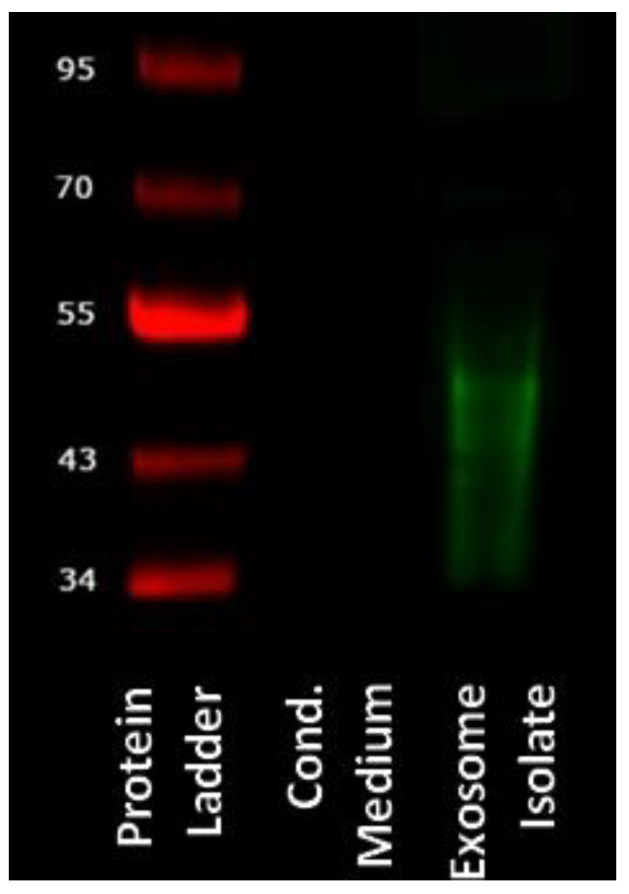
Western blot image. In the left lane is the PageRuler Prestained NIR Protein Ladder. In the right lane, 15 µL of exosome isolate was applied. In the middle lane, the same volume of unconcentrated supernatant was used. The protein ladder is labeled with the respective molecular weights in kDa.

**Figure 2 ijms-25-02190-f002:**
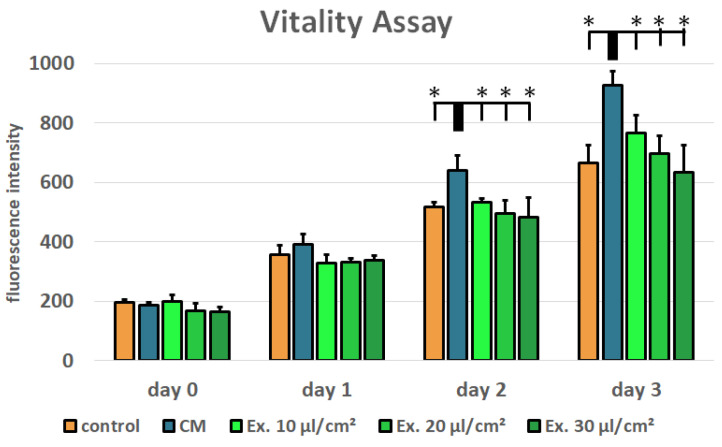
Resazurin assay for cell viability evaluation. Mean values and standard derivations are shown. MCF-7 cells treated with ADSC ES medium show a significantly lower cell viability than cells treated with ADSC-CM (* *p*-value < 0.05).

**Figure 3 ijms-25-02190-f003:**
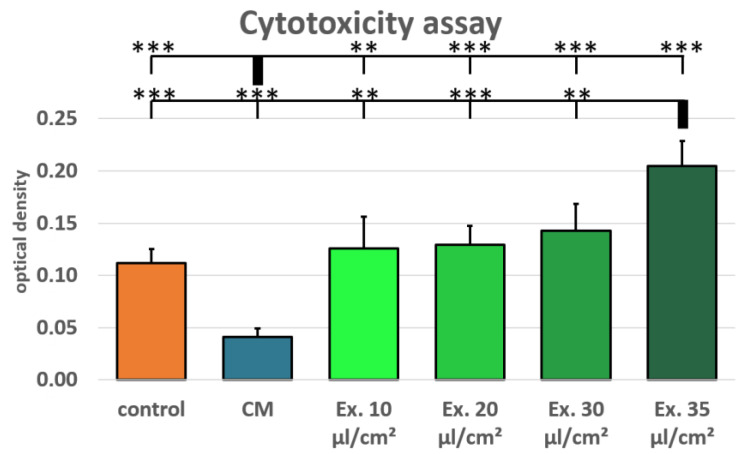
LDH assay for cytotoxicity detection. Mean values and standard derivations are shown. Both the conditioned medium (CM) and the 35 µL/cm^2^ exosome supplementation (Ex. 35 µL/cm^2^) differ highly significantly from all other groups (** *p*-value < 0.01; *** *p*-value < 0.001).

**Figure 4 ijms-25-02190-f004:**
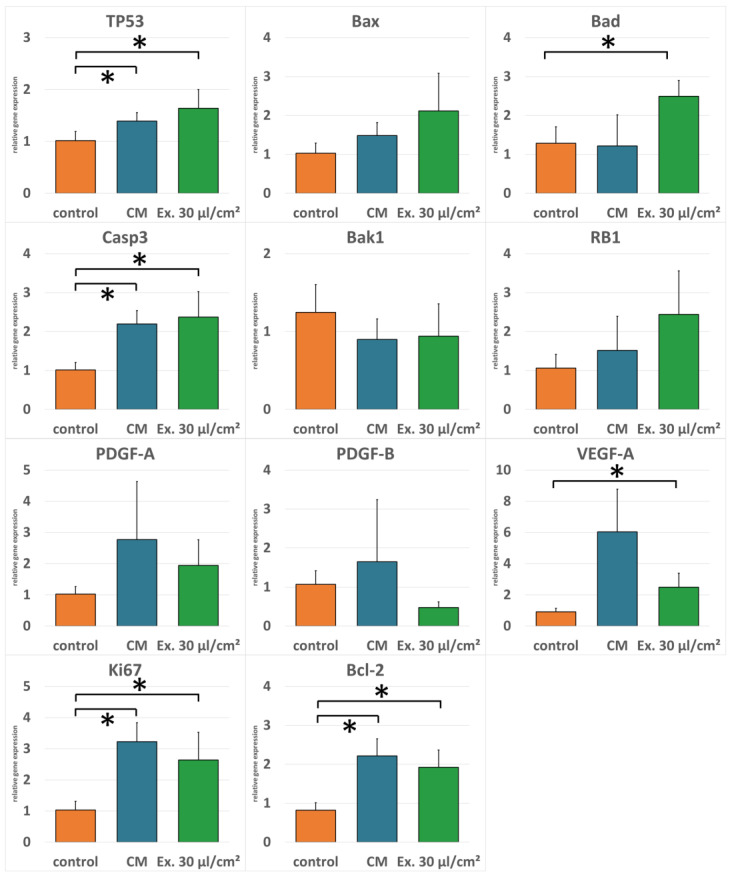
RT-PCR for gene expression. Gene expression of MCF-7 cells after treatment with ADSC-conditioned medium or medium supplemented with 30 µL/cm^2^ of exosome isolate was tested for genes related to apoptosis, angiogenesis, or proliferation. Generally, expression of pro-apoptotic genes was higher in cells treated with the exosome medium compared to the cells treated with the conditioned medium. For anti-apoptotic gene Bcl-2, angiogenesis promoting genes VEGF-A, PDGF-A, and PDGF-B, and the proliferation marker Ki67, expression was higher in cells treated with the conditioned medium compared to the exosome medium. However, not all differences were statistically significant (* *p*-value < 0.05). Mean values and standard derivations are shown.

**Table 1 ijms-25-02190-t001:** Sequences of primers.

Gene	Forward Primer Sequence	Reverse Primer Sequence
Bad	5′-TCACAAATCCTCCCCAAGTGG-3′	5′-GAATGCGCCCTAAATCACTGA-3′
Bak1	data	data
Bax	5′-TTCCGAGTGGCAGCTGAGATGTTT-3′	5′-TGCTGGCAAAGTAGAAGAGGGCAA-3′
Bcl-2	5′-ATCGCCCTGTGGATGACTGAG-3′	5′-CAGCCAGGAGAAATCAAACAGAGG-3′
Casp3	data	data
GAPDH	5′-GGAGCGAGATCCCTCCAAAAT-3′	5′-GGCTGTTGTCATACTTCTCATGG-3′
Ki67	data	data
PDGF-A	5′-CCCCTGCCCATTCGGAGGAAGAG-3′	5′-TTGGCCACCTTGACGCTGCGGTG-3′
PDGF-B	5′-GATCCGCTCCTTTGATGATC-3′	5′-GTCTCACACTTGCATGCCAG-3′
RB1	5′-TTGGATCACAGCGATACAAACTT-3′	5′-AGCGCACGCCAATAAAGACAT-3′
TP53	5′-GGAGTATTTGGATGACAGAAAC-3′	5′-GATTACCACTGGAGTCTTC-3′
VEGF-A	5′-AACCAGCAGAAAGAGGAAAGAGG-3′	5′-CCAAAAGCAGGTCACTCACTTTG-3′

## Data Availability

Data are contained within the article.

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
