# Peer review of "Exosomes from Adipose-Tissue-Derived Stem Cells Induce Proapoptotic Gene Expression in Breast Tumor Cell Line"

_ijms, 2024, doi:10.3390/ijms25042190_

Round 1

Reviewer 1 Report

Comments and Suggestions for Authors

-          How do you make sure that your cells are ADSCs? In my opinion you must contrast that they really are with the traditional protocols and techniques.

-          EVs are confirmed with only one marker. It must be improved with some  more such as CD81, CD9 or flotilin among others. For example some markers related to mesenchymal stem cells or cell death cytotoxicity  markers must improve the characterization of the isolated exosomes.

-          Results:

 --Cell viability: all cell cultures in all medium are continuously growing as time goes by.

-Cytotoxicity: is increased for EV 35-conditioned medium

How do you explain the discrepancy in both results? How is possible to have the ability of increasing cell viability  and at the same time being cytotoxic?

- qPCR: ki67 is overexpressed comparing with control, so cells are growing with Ex 30ul/ml. 

 I am missing the results of PCR for 35ul/cm2

Author Response

Comments and Suggestions for Authors

ANSWER:  We thank the reviewer for the valuable comments that hopefully will help to improve our manuscript. We have tried to address the reviewer’s concerns, however in some cases where new experiments would be needed we will not be able to do this in the time given for the revision. However, in these cases the reviewers’s concern is reflected in the limitations of the study

-How do you make sure that your cells are ADSCs? In my opinion you must contrast that they really are with the traditional protocols and techniques.

ANSWER:  The reviewer is correct that the cell identy needs to be ensured. Our working group has a long history of more than 20 years working with ADSCs and we routinely check our cells for differentiation potential and surface marker expression. However, we really don’t think it is necessary to show nearly identical results that are of little interest to the study’s topic in each and every manuscript. We have added a statement to address this.

-EVs are confirmed with only one marker. It must be improved with some more such as CD81, CD9 or flotilin among others. For example, some markers related to mesenchymal stem cells or cell death cytotoxicity markers must improve the characterization of the isolated exosomes.

ANSWER: We thank the reviewer for the valuable suggestion. However, we will not be able to perform the requested experiments in the time (10 days) given for the revision. We don’t have any exosome isolate left from the experiments, which should be no surprise considering the difficulty and time-consuming nature of harvesting exosomes in greater quantities. However, in our opinion the combination of CD63-positivity and size range is a clear-enough hint that we indeed are working with exosomes. However, we have listed the nature of the exosome confirmation as a limitation of the study.

-Cell viability: all cell cultures in all medium are continuously growing as time goes by. Cytotoxicity: is increased for EV 35-conditioned medium. How do you explain the discrepancy in both results? How is possible to have the ability of increasing cell viability and at the same time being cytotoxic?

ANSWER: EV 35 was tested for cytotoxicity, only, not for cell vitality. The reason for this is, that exosome isolation requires a lot of resources, because a huge amount of cell-culture-conditioned medium is needed to obtain a small amount of concentrated exosomes. For cell vitality several timepoints were evaluated and for RNA isolation a bigger cell culture area is needed, and we never had enough exosome isolate to use the 35-concentration in these settings. Cytotoxicity however can be tested in small wells and we tested a single timepoint, only, allowing us to test a higher concentration. We have already addressed this limitation in the manuscript. Up to the 30 concentration there is no big increase in cytotoxicity to be observed. Therefore, it would not be surprising if cell viability was not decreased. However, and maybe we should have started the rebuttal with this point: cell vitality is clearly NOT rising for the higher exosome concentrations but stagnates at best. Therefore, we do not see nay contradiction in our data.

- qPCR: ki67 is overexpressed comparing with control, so cells are growing with Ex 30ul/ml.  I am missing the results of PCR for 35ul/cm2

ANSWER: As explained in our answer to the previous concern, we were not able to utilize the 35ul/cm2 concentration in the culture plate areas needed for sufficient RNA isolation. We have tried to clarify this limitation by rephrasing the respective sentences.

Reviewer 2 Report

Comments and Suggestions for Authors

1. Summary

In this manuscript, the topic authors discussed is that the exosomes from adipose-dierived stem cells can cause up-regulation of proapoptotic genes in breast tumor cell line. To address this statement, they firstly isolated the exosomes from primary adipose-drerived stem cells, treated MCF-7 cell line with control condition and esomoses, and check for cell viability , cytotoxicity as well as gene expression.  Then RT-PCA was used to check the proapoptotic gene expression.

The general logic of experimental design is clear, but necessary statistical significance tests with method description were missing for gene expression results, as well as the function interpretation and relative references for listed proapoptotic genes. Moreover, some parts of the manuscripts must be improved to make smoother and clearer description, and the figures needs to be improved as well.

In summary, I would suggest to reconsidering this manuscript after the major revision, which is the statistical analysis and the function interpretation.  

2. Detailed comments

Here I only listed the comments for major revision. 

1. Introduction part

1. Page 2, Line 65 : Please add some description and clarification for proapoptotic genes, since it shows this is an important part both in title and results. 

2. Page 2, Line 62~68 : Also please add more conclusive description at the last paragraph of the introduction to clarify why the meaning of the study.

2. Results part

1. Page 2, Line 73, Figure 1: Please mark out the condition of each lane at the x axis.

2. Page 3, Line 83, Line 92, etc: Please add the full name for abbreviations at the first appearance. The listed ES, ADSC-CM are examples, please check for others.

3. Page 3, Line 82~88: Please describe the exact significance test results (including P-value status) both in the figure legend and text.

4. Page 4, Line 108~110: Please add the reference for the proapoptotic gene lists with interpretation. The is because that you need to show the logic and reason why using RT-PCR and check expression of these specific genes. 

5. Page 4, Line 107~116: The statistical significance tests are missing. Please add the significance test results for each comparison and mark out in the figure 4. This is the one of the part that must be improved. 

6. Page 4, Line 117: Please use the vectorgraph to prevent blurry after enlarging. Please also enlarge the font size for the labels of x axis and y axis and use the uniform size for similar parts from different figures. 

7. Page 4, after Line 116: Please add some conclusive description at the differential expression results not only describe the expression change. There is a need to explain what's the biological meaning of theses up/down regulation and why they are important.

3. Materials and Methods part

1. Page 6 to Page 8: Please add the information of applied statistical analysis including methods and tools. This is the one of the part that must be improved. 

2. Page 6, Line 211: What is the meaning of "XXX" in the first sentence?

Author Response

Comments and Suggestions for Authors

In this manuscript, the topic authors discussed is that the exosomes from adipose-derived stem cells can cause up-regulation of proapoptotic genes in breast tumor cell line. To address this statement, they firstly isolated the exosomes from primary adipose-derived stem cells, treated MCF-7 cell line with control condition and exosomes, and check for cell viability, cytotoxicity as well as gene expression. Then RT-PCA was used to check the proapoptotic gene expression.

The general logic of experimental design is clear, but necessary statistical significance tests with method description were missing for gene expression results, as well as the function interpretation and relative references for listed proapoptotic genes. Moreover, some parts of the manuscripts must be improved to make smoother and clearer description, and the figures needs to be improved as well.

 In summary, I would suggest to reconsidering this manuscript after the major revision, which is the statistical analysis and the function interpretation. 

ANSWER: We thank the reviewer for the valuable comments. We have tried to do all changes according to the reviewer’s suggestion which hopefully will improve our manuscript.

  1. Page 2, Line 65: Please add some description and clarification for proapoptotic genes, since it shows this is an important part both in title and results.

ANSWER: We thank the reviewer for the suggestion and agree that we should have given a short description of the genes we have evaluated. We have added some information for the genes, especially for the genes of the Bcl-2 family.

  1. Page 2, Line 62~68: Also, please add more conclusive description at the last paragraph of the introduction to clarify why the meaning of the study.

ANSWER: We thank the reviewer for the suggestion and have extended the description of the meaning of the study a little. However, here and in the previous point we have added text in a manner that the article still reflects the article type “communication”.

  1. Page 2, Line 73, Figure 1: Please mark out the condition of each lane at the x axis.

ANSWER: We have added information about each lane at the bottom of the figure

  1. Page 3, Line 83, Line 92, etc: Please add the full name for abbreviations at the first appearance. The listed ES, ADSC-CM are examples, please check for others.

ANSWER: We apologize for our carelessness and have added “(exosome supplemented)” to the first appearance of “ES” in the line which was originally line 83. However, “CM” in (originally) line 92 is already explained in its first appearance in line 68 (original line counting). We have checked for other abbreviations but only found those that were not further explained intentionally (like LDH, VEGF or PDGF).

  1. Page 3, Line 82~88: Please describe the exact significance test results (including P-value status) both in the figure legend and text.

ANSWER: I am sorry but I am not sure if I fully understood the request. What we are saying here is that on day 2 and 3, the viability values for the conditioned medium (CM) differ significantly from the other 4 treatments on that day, with a p-value below 0.05 (but not below 0.01), and that no other treatments differ significantly from each other. I think this is sufficiently described in both the text and the figure. What else is meant with “exact significance test results (including P-value status)”? However, we have adjusted the figure 2 in a way that the p-value status is now similar to the depiction in figure 3 (which was not commented) and also we have added the p-value status in the text.

  1. Page 4, Line 108~110: Please add the reference for the proapoptotic gene lists with interpretation. The is because that you need to show the logic and reason why using RT-PCR and check expression of these specific genes.

ANSWER: I am not sure how to phrase this, but… we do not have a reference list (and I never heard of the necessity to have one). We wanted to focus on genes of the Bcl-2 family, because their relative expression in regard to each other is an important regulator of apoptosis initiation, and TP53 and Rb1 are apoptosis-related genes known to play important roles in tumor development. Caspase 3 is a protein at the end of the apoptotic cascade and we felt that something would be missing if we would not evaluate one of the caspases, and caspase 3 is well known. We think we have made a good selection of apoptotic markers from different stages of the apoptotic cascade. After all, we cannot show all the genes related to apoptosis, and one can always argue that the selection is arbitrary. However, we haven’t “cherry-picked” the results: we show the results from all the genes that we have tested (with Bak not being in favor of our hypothesis), but there is no predefined list of genes that we utilized or any other reference for that matter.

  1. Page 4, Line 107~116: The statistical significance tests are missing. Please add the significance test results for each comparison and mark out in the figure 4. This is the one of the part that must be improved.

ANSWER: At first, we did not show any significance tests for the PCR results intentionally because we thought that the observed contrary tendencies for proapoptotic genes on the one hand and the antiapoptotic, pro-proliferative and pro-angiogenic genes on the other hand would already support our hypothesis sufficiently, and we anticipated that the observed tendency-differences will not be significant for many genes. However, we thank the reviewer for the suggestion which surely helped to improve the manuscript. We have added the significance tests for figure 4.

  1. Page 4, Line 117: Please use the vectorgraph to prevent blurry after enlarging. Please also enlarge the font size for the labels of x axis and y axis and use the uniform size for similar parts from different figures.

ANSWER: We have used a high resolution image and we don’t see any blurriness after enlargement. However, the reviewer is correct that the font size of the x-axis and y-axis should be enlarged and we have changed figure 4 accordingly. 

  1. Page 4, after Line 116: Please add some conclusive description at the differential expression results not only describe the expression change. There is a need to explain what's the biological meaning of theses up/down regulation and why they are important.

ANSWER: We thank the reviewer for the valuable suggestion and have added some more details to the discussion regarding the biological meaning and hope that our manuscript has improved.

  1. Page 6 to Page 8: Please add the information of applied statistical analysis including methods and tools. This is the one of the part that must be improved.

ANSWER: The reviewer is absolutely correct, of course, and we apologize for our carelessness. Although we initially did not provide statistical tests for the PCR data intentionally, we definitely should have added the information about the statistical tests that we have shown. We have added the information at the end of the manuscript in an extra chapter.

  1. Page 6, Line 211: What is the meaning of "XXX" in the first sentence?

ANSWER: We are sorry, this slipped through our own proofreading. The “XXX” is a placeholder specifically used to NOT forget to insert the correct citation. This is a little embarrassing. We have added the correct citation and removed the “XXX”.

Round 2

Reviewer 2 Report

Comments and Suggestions for Authors

This version of revision is well improved and here are responses to some confusion from authors’ answers. 

For Answer 3: The comments I made here was to suggest to add description for significant results shown in the figure 2 in the text. Since the figure is only the visualization of results, the description is needed to show readers what the result exact means. So the updated part is good now.

Fore Answer 4: This question was based on the manuscript that with no background information for Bcl-2 family. If there is no such information clearly described, there is no smooth logic and would make readers confused why these genes were selected to show expressions. With the description for Bcl-2 family in the background, people can easily understand the whole story.
